# Lycopene: A Critical Review of Digestion, Absorption, Metabolism, and Excretion

**DOI:** 10.3390/antiox10030342

**Published:** 2021-02-25

**Authors:** Joseph Arballo, Jaume Amengual, John W. Erdman

**Affiliations:** 1Division of Nutritional Sciences, University of Illinois Urbana Champaign, Urbana, IL 61801, USA; arballo3@illinois.edu (J.A.); Jaume6@illinois.edu (J.A.); 2Food Science and Human Nutrition, University of Illinois Urbana Champaign, Urbana, IL 61801, USA

**Keywords:** lycopene, absorption, digestion, metabolism, excretion, carotenoid, antioxidant

## Abstract

Lycopene is a non-provitamin A carotenoid that exhibits several health benefits. Epidemiological data support a correlation between lycopene intake and the attenuation of several chronic diseases, including certain types of cancers and cardiovascular diseases. It is currently unknown whether the beneficial effects are from the native structure of lycopene or its metabolic derivatives: lycopenals, lycopenols, and lycopenoic acids. This literature review focuses on the current research on lycopene digestion, absorption, metabolism, and excretion. This review primarily focuses on in vivo studies because of the labile nature and difficulty of studying carotenoids within in vitro experimental models. The studies presented address tissue accumulation of lycopene, the modification of bioavailability due to genetic and dietary factors, and lycopene cleavage by the enzymes ß-carotene oxygenase 1 (BCO1) and ß-carotene oxygenase 2 (BCO2). The current literature suggests that the majority of lycopene is cleaved eccentrically by BCO2, yet further research is needed to probe the enzymatic cleavage activity at the tissue level. Additionally, results indicate that single nucleotide polymorphisms and dietary fat influence lycopene absorption and thus modify its health effects. Further research exploring the metabolism of lycopene, the mechanisms related to its health benefits, and optimal diet composition to increase the bioavailability is required.

## 1. Introduction

Carotenoids are 40-carbon long lipophilic compounds present in many fruits and vegetables, which gives these plants visible colorations ranging from red-pink to yellow [1,2]. There are two classifications of carotenoids based on their chemical structure. Carotenes are purely hydrocarbons that include carotenoids such as ß-carotene and lycopene. Xanthophylls contain oxygen in their chemical structure and include lutein, zeaxanthin, and ß-cryptoxanthin. Carotenoid biosynthesis occurs by the condensation of two C20 hydrocarbon chains to form C40 compounds such as phytoene and phytofluene, which are colorless [3]. Further desaturation of phytofluene generates lycopene, another intermediate of carotenoid biosynthesis. The metabolites of these carotenoids, apocarotenoids, are shorter than the typical 40 carbon structure [4]. The chemical structure of lycopene, C_40_H_56_, contains two unconjugated double bonds and eleven conjugated double bonds and provides plants, such as tomato, watermelon, and guava, their distinctive red color [2,4] (Figure 1). The presence of its conjugated double bonds makes lycopene susceptible to heat-mediated isomerization, which leads to the formation of 9-*cis*, 13-*cis*, or 15-*cis* lycopene isomers, among others [2,3,4,5].

Epidemiological studies show that lycopene consumption as well as serum levels are correlated with a reduced risk for certain types of cancer, cardiovascular disease, and overall mortality [6,7,8]. Given the promising epidemiological data on lycopene, researchers have investigated the bioactivity of lycopene. They have found that it may attenuate various chronic diseases such as cardiovascular disease, non-alcoholic fatty liver disease, oxidative stress, inflammatory pathways, lung cancer, prostate cancer, and male infertility [9,10,11,12].

Although research shows that lycopene can attenuate a wide range of chronic conditions, how lycopene exerts this bioactivity remains relatively unknown. The health effects of lycopene may be related to the antioxidant effects of the native *trans* or *cis* structures and shortened by derivatives such as lycopenals, lycopenols, and lycopenoic acids [2,9,11,13,14]. The studies discussed in this review prioritized in vivo research since during in vitro studies, there can be chemical instability and carotenoid degradation, which can confound findings related to carotenoid bioactivity [15,16,17]. This review focuses upon recent advances in knowledge regarding the absorption, metabolism, and excretion of lycopene while guiding the reader to the relevant research on the potential health aspects of lycopene.

## 2. Drawbacks and Suggestions for Lycopene Research

Tomatoes are the most common lycopene source in the Western diet, and they primarily contain lycopene in its all-*trans* form prior to cooking [2,4,18] (Figure 1). Within uncooked tomatoes, the linear *trans*-form of lycopene aggregates into crystalline structures located within chromoplasts, granting lycopene greater thermal stability against isomerization and degradation [2,4,18,19]. This crystalline aggregation and location in the chromoplast make the all-*trans* form of lycopene poorly bioavailable for mammals [20]. Food processing generally employs various thermal and mechanical techniques that contribute to the destruction of cellular walls and dissociation of lycopene’s crystalline aggregates. These processes allow for lycopene to become more bioavailable during intestinal digestion and absorption [21,22]. On the downside, excess thermal processing leads to isomerization and lycopene oxidation. For example, lycopene is especially unstable when tomatoes are air-dried [19,23]. Therefore, industrial processes alter the amount and isomeric forms of lycopene in tomato products based on the amount and type of heating, length of cooking, and oxygen presence, leading to variable contents and bioavailability in the final product.

When studying carotenoids in vitro, one must consider the importance of carotenoid stability during the length of the assay. One study used prostate cancer cell lines and analyzed ß-carotene retention in the cellular media and found significant degradation for each additional 12 h of the assay [17]. The oxidative degradation and non-metabolic isomerization of carotenoids can affect bioactivity, which can impact the outcomes in these types of studies. We advise researchers to ensure that carotenoids, including lycopene, when used in vitro, are stable during the length of the assay by quantifying these compounds using high-performance liquid chromatography (HPLC).

Two decades ago, Roche Vitamins and Badische Anilin und Soda Fabrik (BASF) developed microencapsulation technologies to deliver nutrients in beadlets [24]. This microencapsulation system resulted in a cost-effective strategy to provide fat-soluble and water-soluble dietary nutrients in a protective food starch or protein matrix. The Roche beadlets were encapsulated with vitamin E and vitamin C as protective antioxidants to ensure stability. Lycopene beadlets largely contain all-*trans* lycopene and some *cis* isomers and can be obtained from BASF and Royal Dutch State Mines (DSM), which now owns Roche Vitamins. The utility and stability of lycopene in beadlets have made them a common lycopene source for murine animal trials [25,26,27,28,29]. The benefits of using this form of lycopene supplementation include delineating the specific impact of lycopene compared to whole foods high in lycopene (e.g., tomatoes), ease of use, and consistent dosage [30,31,32,33]. Limitations with lycopene beadlets are the potentially reduced bioavailability and lack of the whole food matrix associated with lycopene intake from food sources such as tomatoes. Given that the majority of epidemiological findings on the health benefits linked to lycopene are associated with tomato consumption, researchers may encounter contradictory findings between the results obtained in public health studies and experimental studies in which lycopene supplements are provided [34]. The authors suggest that researchers should consider the source and purity of lycopene, as it could explain any conflicting results between studies.

## 3. Influence of Food Matrix Composition on Lycopene Bioavailability

As with most fat-soluble dietary components, lycopene exhibits limited bioavailability from foods [35]. Extensive research has aimed to identify optimal dietary factors to enhance lycopene absorption. Dietary fat enhances carotenoid absorption, which has led to an interest in the interaction of lycopene and fat content in meals [36,37,38,39,40]. In a study by Brown et al., participants consumed equivalent amounts of fresh salads that contained cherry tomatoes (providing 8.6 mg of lycopene) with 0, 6, and 28 g of fat from canola oil [39]. HPLC quantification of the postprandial chylomicron fraction showed a dose-response effect on lycopene absorption. As expected, subjects that consumed salad dressing without fat showed negligible levels of lycopene in this fraction [39]. A study by Goltz et al. compared different fat sources (soy, canola, or butter) and quantities (3, 8, or 20 g) on lycopene absorption [36]. They analyzed lycopene concentration in the triacylglycerol-rich fraction from serum following a single prepared meal containing 5 mg of lycopene from 29 human volunteers. In agreement with Brown et al., these authors found that the highest dose of fat (20 g) resulted in a greater lycopene absorption. They found no significant differences in lycopene absorption from 3 or 8 g of fat, despite Brown et al. finding a significant increase in lycopene absorption for 6 g of fat. The authors stated that the lack of a fat-free control, insufficient statistical power, and a maximal amount of canola oil (20 g compared to Brown et al. 28 g) might have contributed to these differences in carotenoid absorption [36].

Another study by Unlu et al. provided 11 human volunteers a control meal (salsa or salads with no fat) and a carotenoid-rich (40 mg lycopene) meal with different amounts of fat (12 and 24 g) and different fat sources (avocado fruit vs. avocado oil) [38]. They analyzed lycopene content in the triacylglycerol-rich fraction and found that both doses of fat enhanced carotenoid absorption ~4.4 times more than the fat-free control diet, but neither dose nor source exhibited greater carotenoid absorption over the other [38]. A study by White and colleagues enrolled 12 women that were given vegetable salads containing 0, 2, 4, 8, and 32 g of soybean oil and 4.5 mg of lycopene. Blood was taken after fasting and 2–9.5 h post-ingestion of the meal [37]. They analyzed carotenoid content in plasma chylomicron fraction, and a linear relationship was established between lycopene absorption and 0–32 g of soybean oil. The possible reason for these conflicting results is the use of linear mixed-effects regression modeling, which predicted the lycopene values against the soybean oil dosing effect, rather than quantifying by area under the curve (AUC) or maximal concentration (C_max_).

The prior mentioned study by Goltz et al. showed that varying the source of fat (soybean oil, canola oil, and butter) did not alter lycopene absorption, and only the amount of ingested lipid increased lycopene absorption [36]. These results are consistent with the study by Lee et al. that enrolled six human subjects to test whether dietary fat sources can modify lycopene absorption [40]. During the first feeding period, participants were given 200 g of tomato soup (totaling 33 mg of lycopene) and 230 g of canned tomatoes (totaling 13 mg of lycopene) with 20 mL of olive oil daily for 7 days. After a 3 week washout period, subjects repeated the test with sunflower oil. HPLC analyses failed to find the differences in lycopene plasma levels, suggesting that fatty acid composition does not alter lycopene levels [40].

These studies point toward a minimum threshold of fat that needs to be consumed to elicit an increase in lycopene absorption for differing food matrices. The described studies used fresh, unprocessed tomato products, which would warrant a higher fat intake for adequate lycopene absorption compared to processed tomato products due to differences in bioavailability. Apparently, the type of fat is not important. Results from these and other studies suggest that optimal lycopene absorption requires a minimum of 10 g of fat in a meal containing processed tomato products, while 15 g (just over 1 tablespoon) may be necessary for uncooked foods like salads or raw tomatoes [36,37,38,39,40].

## 4. The Impact of Lycopene Isomeric Profile on Its Intestinal Absorption

Early publications suggested preferential absorption for *cis*-isomers of lycopene compared to the all-*trans* configuration. This assumption was based on studies showing that subjects consuming tomato juice showed a relative increase of 9-*cis* lycopene serum levels despite the relatively minor amounts present in the tomato juice [22,41,42,43]. These reports were supported by our own findings in ferrets, a model widely used to study carotenoid accumulation [44]. We observed preferential accumulation of *cis*-form lycopene into serum and tissues from ferrets given a single oral dose of Lycored™, a lycopene tomato extract (40 mg of lycopene per kg of body weight). These results led us to the assumption that *cis* isomers are taken up more effectively into lipid micelles than their *trans* counterparts [43]. The exact mechanism(s) that lead to an enhanced *cis* isomeric profile in human serum and tissue remain elusive, but preferential absorption of *cis* isomers, in situ heat-induced isomerization, or enzymatic isomerization that occurs within body tissues could all be potential explanations [22,41,42,43].

The development of in vitro methods that produced adequate amounts of isotopically-labeled carotenoids from tomato cell culture allowed for tracer studies on lycopene absorption and metabolism to be conducted [45,46,47,48]. Moran and colleagues supplemented eight volunteers with a single dose of a ^13^C-labeled lycopene mixture containing 10.2 mg of 82% all-*trans* lycopene, which was purified from tomato cell culture [48]. Researchers collected plasma hourly for 10 h and at 1, 3, and 28 days after dosing. Their results showed that the absorption of *cis* and all-*trans* did not significantly differ (~24%), contrary to prior thinking [22,41,42,43]. Instead, their data demonstrated that lycopene underwent post-absorption isomerization from *trans*- to *cis*-isomers. The use of compartmentalization modeling estimated that the rate of in vivo all-*trans*-to-*cis* lycopene isomerization to be ~58% of the fast-turnover pool per day, but the exact mechanism for this isomerization could not be ascertained from this study [48]. A possible explanation is *trans*-to-*cis* isomerization occurs by heat induction within body tissues [4,48,49]. While no mammalian lycopene isomerase has been discovered to date, plants contain numerous carotenoid isomerases [50]. Therefore, such an enzyme may remain to be characterized; otherwise, we must conclude that *cis* isomers of lycopene in mammalian tissues result from differential absorption from dietary sources, transport, degradation, and as a result of heat-induced isomerization within the body tissues.

## 5. Lycopene Cleavage in Mammals: The Role of ß-Carotene Oxygenase 1 (BCO1) and ß-Carotene Oxygenase 2 (BCO2)

ß-carotene oxygenase 1 (BCO1) and BCO2 are iron-dependent enzymes responsible for carotenoid cleavage in mammals [51]. BCO1 is present in cytosol and cleaves ß-carotene and apocarotenoids centrically [13]. BCO2 is present in the mitochondria and cleaves a wide variety of carotenoids and apocarotenoids eccentrically [52].

Single nucleotide polymorphisms (SNP) impact the catalytic activity of BCO1, as demonstrated by the association between carotenoid circulating levels and SNPs located in the coding region for BCO1 [53,54]. BCO2 activity in humans remains relatively unstudied, with some research suggesting BCO2 to be enzymatically inert in the retina. A recent paper published challenges these results, showing that human BCO2 is expressed as a preprotein with a mitochondrial targeting sequence [55]. The removal of this sequence allowed for enzymatic activity in vitro. Additionally, the insertion of a primate-specific BCO2 amino acid group into murine BCO2 did not hamper the enzymatic activity. These studies suggest that both enzymes are active in humans, contributing to carotenoid metabolism. The interest in these enzymes led to the development of mutant mice with the ablation of BCO1 (*Bco1*^−/−^) or BCO2 (*Bco2*^−/−^), allowing for the study of the interaction of BCO1 and BCO2 with specific carotenoids.

Using ß-carotene as an example of the metabolic actions of BCO1 and BCO2, BCO1 centrally cleaves ß-carotene at the 15,15′ double bond, which produces two molecules of retinaldehyde [56]. The retinaldehyde molecule can be reduced to retinol or oxidized to retinoic acid, which can be oxidized further by cytochrome p450 (CYP450) enzymes [56]. BCO2 metabolism of ß-carotene results in cleavage at the 9′10′ double bond [56]. This cleavage produces the metabolites ß-apo-10′-carotenal and the volatile compound, ß-ionone. Therefore, BCO2 cleavage of ß-carotene can only form a single molecule of retinaldehyde upon further action of BCO1 [57].

Animal trials have been carried out to elucidate which cleaving enzyme is responsible for lycopene cleavage or whether other metabolic pathways are involved in this process. One murine study conducted by our group utilized wild-type, *Bco1*^−/−^ and *Bco2*^−/−^ mice fed a tomato powder, lycopene from beadlets, and control diet (97 mg per kg, 81 mg per kg, and 0 mg diet of lycopene, respectively) [58]. After 30 days of feeding, total lycopene increased in the hepatic tissue of *Bco2*^−/−^ mice and was lower in the livers of *Bco1*^−/−^ mice compared to those from wild-type. *Bco1*^−/−^ mice showed an upregulation of *Bco2* in adipose tissue, but lycopene levels remained unchanged compared to the wild-type mice group. In contrast, *Bco2*^−/−^ mice accumulated more lycopene in the adipose tissue than the other experimental groups, suggesting preferential cleavage of lycopene by *Bco2*.

These findings are further supported by another animal trial where 3-week-old *Bco2*^−/−^ and wild-type mice were fed a control, a tomato powder diet, or a lycopene beadlet-containing diet for 3 weeks (0 mg, 384 mg per kg and 462 mg per kg of diet, respectively) [28]. This study aimed to analyze the effects of tomato or lycopene feeding on the expression of *Bco1*, along with analyzing tissue lycopene and its metabolic derivatives. *Bco2*^−/−^ mice had increased hepatic *Bco1* expression irrespective of the diet compared to wild-type mice [28]. Additionally, *Bco2*^−/−^ mice had higher serum and hepatic concentrations of lycopene compared to wild-types. The lycopene derivatives, apo-12′-, apo-8′-, and apo-6′- lycopenals were found in diets and in the livers of *Bco2*^−/−^ mice fed either a lycopene or tomato diet. These findings suggest that either diet or tissue metabolism could be sources of lycopene derivatives.

Another animal trial by our group investigated whether the presence of BCO2 altered the effects of lycopene or dietary tomato on prostate carcinogenesis [59]. At weaning, transgenic adenocarcinoma of the mouse prostate (TRAMP)/*Bco2*^+/+^ mutant mice and TRAMP/*Bco2*^−/−^ mutant mice were fed an AIN-93G control or lycopene and tomato treatment diets containing 0 mg, 384 mg per kg, and 462 mg per kg of diet, respectively. Lycopene was mixed in the diet in the form of beadlets or as tomato powder (10% total *w*/*w*) for a total of 15 weeks. We found that the inhibitory effects of tomato and lycopene on prostate cancer occurrence and lesion development were attenuated in *Bco2*^−/−^ mice, strongly suggesting that BCO2-derived lycopene metabolites reduce prostate cancer development and progression. In agreement with other studies, TRAMP/*Bco2*^−/−^ mice had increased serum lycopene levels compared to animals with intact *Bco1*.

These animal trials show a preference for relative BCO2 enzymatic activity on lycopene in vivo [28,58]. The presence of BCO2 inhibits lycopene accumulation in several murine animal trials, and the ablation of BCO2 results in increased lycopene levels in plasma and tissues compared to the wild-type mice [28,58]. The presence of lycopene metabolites in tissues, even with ablation of BCO2, indicates that other pathways may be involved in lycopene metabolism, although to lesser degrees. Future research studies should probe the enzymatic activity of lycopene cleavage at the individual tissue level.

## 6. Digestion and Tissue Distribution of Lycopene

Mastication and peristalsis are essential for lycopene bioavailability because of the mechanical disruption of food releasing lycopene from the food matrix [60,61]. Once lycopene has reached the stomach, the mechanical churning and action of enzymes and acid partially release carotenoids from the food matrix. Lycopene can then be internalized into the lipid droplets and released into the small intestine, where enzymes and bile acids continue the breakdown of the food matrix [60]. Lycopene can then be incorporated into the lipid micelles and be taken up by enterocytes [60,62]. Current evidence suggests that lycopene may be absorbed by passive diffusion and by the action of scavenger receptor class B type 1 (SR-B1), which is known to mediate the absorption of other carotenoids such as ß-carotene and lutein [7,29,62]. BCO1 and BCO2 are highly expressed in the intestine, suggesting that lycopene may be partially cleaved in the enterocyte [63]. However, it is assumed that the majority of lycopene is packed into chylomicrons in its intact form and transported to the lymphatic system [1,64,65]. This process could be mediated by the microsomal triglyceride transfer protein (MTTP), an enzyme that delivers triglycerides and other lipids into nascent chylomicrons [66]. The enterocytes release chylomicrons into the lymph and then into the portal circulation, where extrahepatic lipoprotein lipases can partially degrade them into chylomicron remnants. There appears to be some passive diffusion of lycopene into cells during chylomicron degradation before clearance into the liver [67].

The liver clears chylomicron remnants present in the portal system, and lycopene and perhaps some of its metabolites are packaged into very-low-density-lipoproteins to be transported into the bloodstream [67]. Carotenoids are exchanged between lipoprotein classes, but lycopene is primarily carried by low-density lipoprotein (LDL) [68]. Tissues take up lycopene from lipoproteins by interacting with certain membrane proteins such as SR-B1 and CD36 [29,69]. Lycopene preferentially accumulates in the liver, but it can be found in other organs, including the adipose tissue, adrenal glands, testes, ovaries, kidneys, lungs, skin, and the prostate [4,49,70] (Figure 2).

## 7. Intestine Specific Homeobox (ISX) and Scavenger Receptor Class B Member 1 (SR-B1): Role as Potential Influencers of Lycopene Tissue Absorption

ISX is a transcription factor that regulates the expression of several genes, including SR-B1 and BCO1 [71]. Lobo and colleagues uncovered an interaction between ß-carotene and the transcription factor intestine-specific homeobox (ISX) [72]. They found that ISX expression, a retinoic acid target gene, was upregulated following ß-carotene consumption. As a result, ISX suppresses the expression of SR-B1 and BCO1 in the intestine, reducing carotenoid uptake and cleavage, respectively [72]. Further research on this topic showed that *Isx*^−/−^ mice also exhibited increased xanthophyll and alpha-tocopherol absorption due to SR-B1 upregulation in the enterocyte [73].

While these studies did not investigate whether ISX and SR-B1 regulates lycopene absorption, several genome-wide association studies (GWAS) have reported that SNPs in the SR-B1 gene (SCARB1) were associated with lower lycopene serum concentrations [74,75]. A study by Moussa et al. provides further mechanistic evidence that SR-B1 partially facilitates lycopene uptake into human intestinal cells and shows increased lycopene serum levels in a murine animal model with overexpression of SR-B1 [29].

Since it has been shown that β-carotene has an inhibitory effect on carotenoid absorption due to its effect on ISX expression [72,73,76], it is essential to investigate whether these findings apply to lycopene. One would imagine that lycopene absorption should not affect retinoic acid signaling in the enterocyte to the same extent as retinoids derived from ß-carotene. Theoretically, vitamin A deficiency would increase the expression of SR-B1 and BCO1, which would facilitate an increase in lycopene absorption and possibly favor the central cleavage by BCO1 rather than by BCO2 [74,75].

## 8. Genetic Polymorphisms Impact on Bioavailability and Distribution of Lycopene

There has been a growing interest in the physiological effects of genetic variants in key transporters and enzymes that impact carotenoid absorption and cleavage. While genetic mutations that inactivate BCO1 and BCO2 in humans have not been found to date, mutations in other animals have been detected. For example, mutations in BCO2 lead to an increased presence of ß-carotene in milk and adipose tissue of cows [77]. These animals showed reduced hepatic stores of vitamin A, and DNA analyses revealed the presence of a stop codon in the BCO2 coding region [77].

While BCO2 SNPs have been detected in humans, the current data show that they have little impact on lycopene absorption [66]. A research study by Borel et al., discussed further below, investigated postprandial lycopene concentrations from 33 humans and found no interaction between BCO2 SNPs and lycopene bioavailability.

BCO1 SNPs have been reported to modulate lycopene concentrations in humans in several GWAS studies [53,78]. Among the different BCO1 variants, SNP rs6564851 is likely the most relevant. This SNP was correlated with higher ß-carotene levels and lower lycopene, lutein, and zeaxanthin concentrations in the serum [53].

Our group recently showed that changes in rs6564851, which affects BCO1 activity [79], correlated with lower total and non-high-density lipoprotein (non-HDL) plasma cholesterol levels [80]. Regarding lycopene, a study evaluated a group of prostate cancer patients that were instructed to drink 0, 1, or 2 cans of tomato-soy juice daily for 24 days, providing 20.6 mg of lycopene per can. Blood samples were collected before and at the end of the study. During prostatectomy, prostate tissues were collected for genetic and carotenoid analyses. Their results showed that men with either BCO1 SNPs, rs6564851 or rs12934922, had increased lycopene concentration in serum and prostate tissue independent of the diet given [78]. These results suggest that specific SNPs of BCO1 are associated with more effective lycopene uptake in humans.

There have been research efforts to evaluate SNPs within other genes that may be pertinent to lycopene absorption [66,74,75]. For example, Borel et al. investigated SNPs from genes that may impact lycopene levels, such as those tied to absorption, catabolism, tissue uptake, and oxidation in 33 people [66]. Saliva samples and blood serum were collected for analysis. After a 48-h washout period with no consumption of lycopene-containing foods and an overnight fast, subjects were fed tomato puree (9.7 mg of lycopene) within similar times to ensure consistent gastric emptying. They found a 70% interindividual variability of chylomicron lycopene levels between subjects and found that 72% of that variance could be accounted for by SNPs investigated in this study, suggesting that genetic factors influence lycopene absorption. Additionally, several SNPs within the elongation of very long fatty acid protein 2, MTTP, adenosine triphosphate-binding cassette subfamily b member1, and superoxide dismutase 2 genes could be the candidates for future mechanistic research on lycopene absorption and metabolism [66].

## 9. Lycopene and Metabolites: Bioactive or Merely Present?

Initial lycopene catabolism results in the production of lycopenoids (Figure 3). The action of additional enzymes is expected to produce a variety of additional metabolites. Lycopene metabolites have been detected in human breastmilk and serum at micro- or picomolar levels [81,82]. Unfortunately, most research evaluating the metabolic and potential health benefits of lycopenoids has been carried after dosing with pharmacological levels of lycopene or lycopenoids.

Research done by Ip et. al. showed that pharmacological doses (10 mg per kg diet) of the putative lycopene metabolite APO-10′-lycopenoic acid (APO10) attenuates glucose intolerance and reduces hepatic inflammation in mice fed a high-fat diet [83]. Their data showed that APO10 reduced the expression of several inflammatory markers and Sirtuin 1 (SIRT-1), a nicotinamide adenosine dinucleotide-dependent deacetylase that is associated with energy and hepatic lipid homeostasis [84]. This led to the attenuation of liver inflammation and the delay of liver tumorigenesis [83]. Additionally, in vitro studies showed that APO10 interacted with retinoic acid receptor beta (RARß) promoter activating the expression of RARß, providing evidence toward APO10 being a retinoic acid analog [85].

Research by Gouranton et al. supports this work by showing that APO10 may interact with RARs and attenuate inflammation by reducing the pro-inflammatory cytokines interleukin-6 and interleukin-1β in mice [86]. APO10 also upregulated cytochrome P450 26A1 and RARß within mouse adipose tissue, two well-known retinoic acid-responsive genes [87]. These findings support the role of APO10 as a retinoic acid analog [83,86]. One study fed mice a liquid high-fat diet (60% kcals from fat) supplemented with APO10 (40 mg per kg diet) for 16 weeks and found that APO10 upregulated the expression of SIRT-1, which could decrease fat accumulation and protect animals from hepatic steatosis [9]. Supporting this research, another study using *Bco2*^−/−^ mice supplemented for 12 weeks with either APO10 (10 mg per kg diet) or lycopene (100 mg per kg diet) showed that APO10 also upregulated SIRT-1 [11]. Additionally, they found that lycopene modestly upregulated peroxisome proliferator-activated receptors α and γ, which have the potential to regulate fatty acid catabolism and reduce lipid accumulation in the liver. This study supports the notion that native lycopene and its potential derivatives attenuated hepatic steatosis progression and occurrence through different pathways [11]. Research on the other putative lycopene metabolites, such as apo-14′-lycopenoic acid, apo-8′-lycopenal, apo-10′-lycopenal, and apo-12′-lycopenal, exhibited bioactivity in vitro [88,89,90].

Gouranton and Ip’s studies suggest that lycopenoids can increase the expression of RARß, a retinoic acid target gene, which indicates that these compounds bind to RARs to influence gene expression. These findings suggest that lycopenoids may regulate gene expression by binding to similar nuclear receptors retinoids that retinoic acid binds too. One study aimed to investigate this question, using in vitro and computational methods to determine whether apo-13-lycopenone and apo-15-lycopenal might function as retinoid antagonists [91]. The results indicated that apo-13-lycopenone and apo-15-lycopenal readily dock onto RARα, which may allow them to function as antagonists to retinoic acid. However, neither apo-13-lycopenone nor apo-15-lycopenal has been reported to be found in human plasma or tissues [91]. The short half-life, very low serum concentrations, and the highly reactive nature of these compounds would make them difficult to detect. Additional systematic studies are needed with physiological concentrations of known lycopenoids to determine their potential impact on gene expression.

Few studies have been conducted on the transportation and tissue concentration of lycopenoids in vivo. One study (discussed further below) using men fed ^14^C-labeled lycopene showed that polar metabolites were excreted in the urine. [92] In other work, seven human participants drank a soy-fortified tomato juice (21.9 mg of lycopene) for eight weeks [82]. All serum samples contained apo-6′, 8′-, 10′-, 12-, and 14′-lycopenal, but the levels were 1000 times lower than lycopene. Additionally, they found that these lycopene derivatives were found within the food consumed, which shows that some lycopenoids can be obtained from the diet [82].

Animal studies performed in ferret and rat models have detected lycopene metabolites following lycopene or tomato feeding. One study from our group utilized F344 male rats fed 25 mg lycopene per kg diet for 30 days. They were given an oral dose of ^14^C-labeled lycopene after their feeding trial and taken down either at 5 or 24 h post-dose. Small amounts of apo-8′-lycopenal and apo-12′-lycopenal were identified within these rat livers [93]. When ferrets were dosed with 10% dry powdered lycopene dissolved in corn oil daily for nine weeks, apo-10′-lycopenol but not APO10, nor apo-10′-lycopenal was detected within the lungs [94]. The metabolite of lycopene, apo-10′-lycopenal, is likely an intermediate compound that could theoretically be either reduced to apo-10′-lycopenol or oxidized to APO10 [94]. To support this hypothesis, apo-10′-lycopenal was added to ferret liver homogenates with or without nicotinamide adenine dinucleotide (NAD+ or NADH). The authors provided in vitro evidence that NAD+ and NADH were needed as cofactors to produce APO10 and apo-10′-lycopenol/APO10 respectively, although these findings should be confirmed in vivo [94]. Considering the instability and reactivity of aldehyde and acid forms and that alcohol-substituted apocarotenoids can be esterified by the lecithin: retinol acyltransferase, it is not surprising that the alcohol form of lycopene cleavage products was only found in the lungs, where this enzyme is highly expressed [57].

While the current literature suggests that lycopene metabolites (derived from the diet or post-absorption metabolism of lycopene) are present in low levels within human serum, breastmilk, and urine, several gaps remain to be addressed [81,82,92]. Some researchers report the biological activity of lycopene metabolites, but the amount used in these studies would be considered pharmacological [83,85,86]. Relative to intact lycopene in the diet, these metabolites were found to be 1000 times less concentrated [82]. For example, in order for 23 mg of lycopene derivatives (apo-6′, 8′-, 10′-, 12-, and 14′-lycopenal) to be consumed from the diet, one would need to consume close to 40 kg of tomato paste [22,82]. Additionally, it is unknown which tissues metabolize lycopene or accumulate its metabolites, how these metabolites are transported in vivo, and the pathways responsible for the secondary metabolism of the lycopenoids.

## 10. Excretion

Isotopically labeled carotenoids are an effective way of investigating lycopene excretion in vivo. Such studies have been carried out using lycopene with rats, gerbils, and humans [92,95,96,97]. One study used F344 male rats that were fed an AIN-93G diet enriched with lycopene (25 mg per kg diet) [95]. The rats were anesthetized and gavaged with a radioactive dose of 98% all-*trans*
^14^C-labeled lycopene (0.246 mg). The rats were placed in metabolic cages to collect urine and feces. Within the feces, 36% of the ^14^C radio label from the lycopene dose was recovered, and 1% of the dose was recovered in urine [95]. After one week, 68–72% of ^14^C radiolabel from lycopene was eliminated in feces. Most of this excretion (54–58%) occurred during the first 48 h [95].

Another study used Mongolian gerbils fed either a control diet or a diet containing 10% tomato powder (297 mg lycopene per kg diet) [96]. After 26 days, the gerbils (n = 2) were anesthetized with isoflurane and dosed with ^14^C-radiolabeled lycopene (1.31 mg) in cottonseed oil [96]. After 24 h, 42.3% of the dose of ^14^C lycopene was excreted in the feces. [96] Both rodent models demonstrated that much of the dosed lycopene is unabsorbed and generally is found in the feces within the first 48 h.

Ross et al. dosed male volunteers with 10 mg of ^14^C-radiolabeled lycopene and collected urine at 4 and 12 h post-treatment [92]. The urine was then collected on the 13th, 21th, 28th, 35th, and 42nd day after the initial lycopene dose. They reported that 17–19% of ^14^C labeled lycopene dose was excreted through the urine after the first 24 h. Additionally, this group detected polar metabolites of lycopene within the urine, confirming that mammals eliminate lycopene partially as polar derivatives. Moran and colleagues found that in eight subjects, a range of 17–29% of radiolabeled lycopene was absorbed [48]. This was lower than previous research, which found a 34% lycopene absorption from a tomato-paste drink possibly due to greater accuracy of the tracer method used, dose formulation differences, and the inclusion of unlabeled lycopene in the compartmentalization model [47,48]. As reviewed above, Moran found a higher turnover rate of absorbed all-*trans* lycopene compared to *cis*- lycopene, suggesting that there may be differential degradation or metabolism of the different isomers within body tissues and that *trans*–to–*cis* isomerization may occur. Further analyses of serum samples from Moran et al. were performed by Cichon et al. and a 1,2 lycopene epoxide was found in plasma beginning 2–4 h post-dosing and represented 1.7% of the dose [97]. This work supports previous work by Khachik et al., who enrolled three lactating women one month postpartum and analyzed breastmilk and serum for carotenoids [98]. This group detected 2,6-cyclolycopene 1,5-diol I and 2,6-cyclolycopene-1,5-diol II and hypothesized that these would be the hydrolysis products of lycopene epoxides. While Cichon et al. detected other lycopene epoxide derivatives, these particular lycopene diols were not detected.

The results from preclinical and clinical tracer studies investigating lycopene excretion show substantial gaps in knowledge. While the rodent animal models allow for a controlled environment and the ability to extract tissues, there are significant physiological differences in rodents that may influence the conclusions made. Notably, most rodents do not absorb carotenoids as effectively as humans [44]. Within the human tracer studies, there are limited numbers of subjects and collection points [48,92]. Studies quantifying lycopene metabolic products in plasma have had difficulty identifying metabolic products of lycopene. Further research is warranted to reveal enzymatic pathways responsible for the production of lycopene metabolites, such as epoxides and diols, in vivo.

## 11. Conclusions

Lycopene, β-carotene, and lutein are the most abundant carotenoids found in human serum and tissues. Epidemiological data suggest that lycopene is associated with a reduced risk of an array of chronic conditions, which warrants further investigation into the bioactivity of this carotenoid [6,7].

Despite the large body of literature on lycopene, it has been challenging to assess the mechanism of action in vivo. The labile nature of lycopene can expose it to isomerization and oxidative degradation, which can contribute to contradictory results coming from in vitro research or even ex vivo analyses [16,19,20]. While useful for controlled settings and genetic knockouts, murine animal models do not absorb carotenoids effectively as humans do, complicating research analyzing absorption, transportation, and excretion mechanisms. Human studies experience high inter-individual variability in lycopene levels, attributed in part to food preparation procedures, differential bioavailability depending on the presence of co-consumed dietary fat, and differences in genetic polymorphisms associated with lycopene absorption and metabolism [36,37,38,53,54,66,78]. Lastly, lycopene metabolites have proven difficult to study due to their low tissue concentrations before being further metabolized by CYP450 or other enzymes.

It is unclear how lycopene or its metabolites produce the health benefits reported from epidemiological association studies. Some research suggests that lycopene derivatives activate a wide range of biological processes that can attenuate chronic conditions, but the concentrations utilized in those studies are often at pharmacological levels [83,85,86]. The physiological concentrations of these metabolites are much lower than the doses in these studies, questioning whether these metabolites exert this bioactivity in the amount found in humans serum and tissues [83,85,86]. For researchers to address these gaps of knowledge, they should focus on questions related to lycopene cleavage products, how they are transported, accumulated, and degraded, and most importantly, whether lycopene and its metabolites impact other metabolic pathways such as inflammation and cell survival (See Table 1 for a partial list of current gaps in knowledge).

## Figures and Tables

**Figure 1 antioxidants-10-00342-f001:**
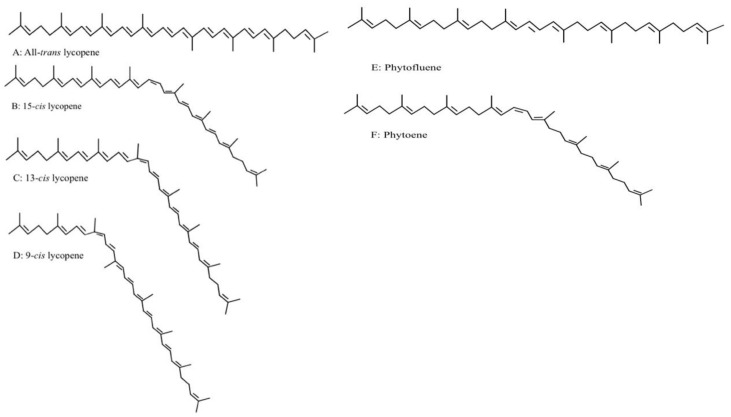
Chemical structure of lycopene, lycopene isomers (15-*cis*, 13-*cis*, 9-*cis*), phytofluene, and phytoene.

**Figure 2 antioxidants-10-00342-f002:**
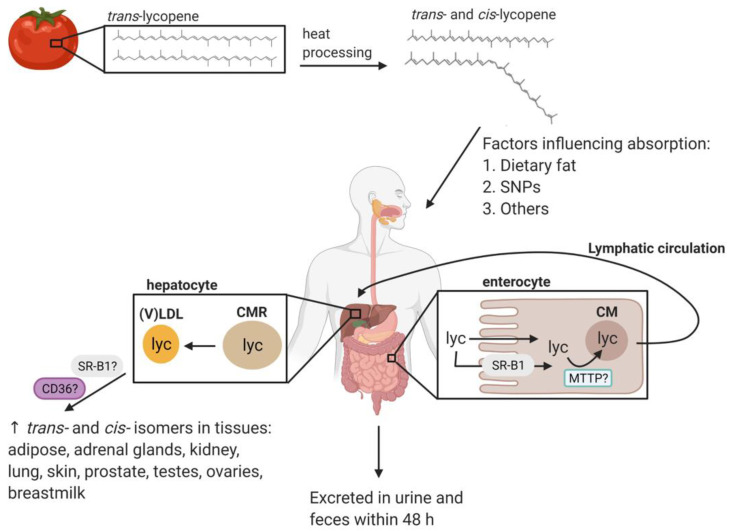
Lycopene (lyc) is present in plant (e.g., tomato) chloroplasts in microcrystalline structures formed by the linear, *trans*-isomer of lyc. Heat processing leads to increased lyc bioavailability as well as lyc isomerization, creating a mixture of *trans*- and *cis*-isomers found in cooked tomato products. The absorption of the fat-soluble carotenoid, lyc, is enhanced when consumed with a meal containing dietary fat (about 10 g dietary fat with cooked tomatoes or about 15 g dietary fat with uncooked tomatoes). Other factors such as single nucleotide polymorphisms (SNPs) are associated with lycopene absorption and metabolism although more definitive work is needed to increase our understanding of the interaction of SNP’s and lycopene metabolism in various tissues. In the intestines, *trans*- and *cis*-isomers of lyc may enter enterocytes either by passive diffusion or facilitated by scavenger receptor class B type 1 (SR-B1). *Trans*- and *cis*-lyc are incorporated into chylomicrons (CM), possibly facilitated by the activity of microsomal triglyceride transfer protein (MTTP), and transported to the liver via lymphatic circulation, during which some lyc may passively diffuse into tissues. In hepatocytes, contents of CM remnants (CMR) are repackaged into very low-density lipoproteins (VLDL) and distributed to the tissues via the bloodstream. Lipoproteins, especially low-density lipoprotein (LDL), interact with the tissues to transfer lyc, a process thought to be mediated in part by cell surface proteins SR-B1 and CD36. Although lyc preferentially accumulates within the liver, stores are also found in other tissues such as adipose, adrenal glands, kidneys, lungs, skin, prostate, testes, ovaries, and breastmilk. Dietary lyc is not efficiently absorbed. One study showed that an average of ~24% of a purified lyc dose was absorbed by human subjects [48].

**Figure 3 antioxidants-10-00342-f003:**
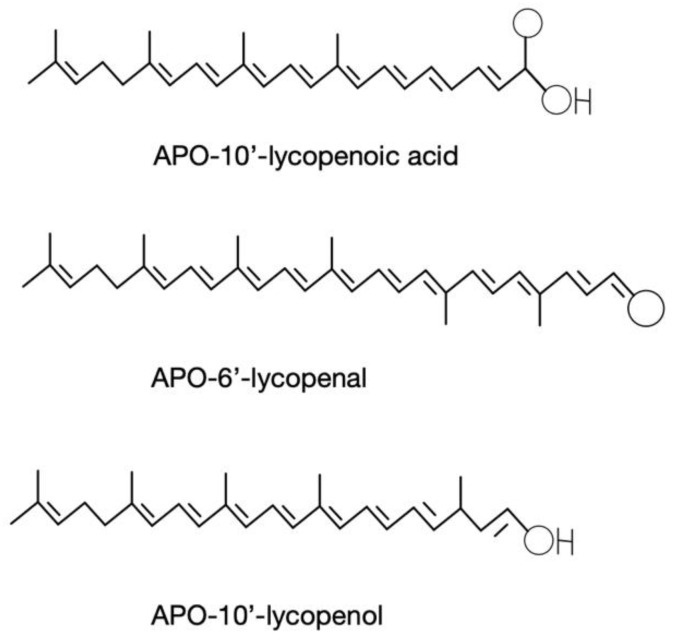
Examples of lycopene metabolites.

**Table 1 antioxidants-10-00342-t001:** Gaps in knowledge regarding lycopene digestion, absorption, metabolism, excretion, and bioactivity.

**Digestion and Absorption**
What are the functional SNPs associated with lycopene absorption? How important are these variants?
Does the ß-carotene/ISX/SR-B1 pathway influence the absorption of lycopene?
How well are lycopene metabolites absorbed?
Which are the amounts or types of fat for an optimal lycopene absorption?
Other than fat, what other dietary components impact lycopene absorption?
**Metabolism**
What enzyme(s) preferentially cleave lycopene?
What tissue(s) metabolize lycopene?
Which lycopene metabolites are present in human tissues?
Is lycopene isomerized within human tissues?
Are other pathways involved in lycopene catabolism? To what extent and how do they interplay with BCO1/2?
What are the enzymatic pathways for secondary metabolism of lycopenoids?
**Excretion**
What pathways and tissues regulate lycopene excretion in humans?
What enzymes metabolize lycopenoids?
**Bioactivity**
Are the antioxidant properties of lycopene responsible for some, if not all, its health benefits?
Do lycopene metabolites have bioactivity at physiological concentrations?
Do both intact lycopene and lycopenoids have bioactivity in humans?
How do interindividual differences affect the bioactivity of lycopene or its metabolites?

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
