# Peer review of "Lycopene: A Critical Review of Digestion, Absorption, Metabolism, and Excretion"

_antioxidants, 2021, doi:10.3390/antiox10030342_

Round 1

Reviewer 1 Report

The manuscript by Arballlo et al. summarizes the recent knowledge about lycopene metabolism.

The manuscript is well written, but the clarity could be further improved by:

  • addition of a Table summarizing the studies discussed in the subchapters
  • implementation of the BCOs in Figure 2 not only in relation to the SPNs, e.g. in which organs BCOs are  functional in humans
  • some linguistic improvement in Table 1

Author Response

The 3rd reviewer suggested three items for consideration by the authors.  The edits can be found highlighted in yellow in the attached revision.  Below are the replies to the three suggestions.
1. The addition of a Table summarizing the studies discussed in subchapters.  
We respectively decline to create such a table as it would almost as long and more bulky than the original manuscript.

2. Implementation of the BCOs in Figure 2 not only in relation to the SNPs, e.g. in which organs BCOs are functional in humans.
Initially we thought that this was an excellent idea and then realized, after checking the literature, that not enough is known about tissue specific expression and activity of BCO1 or 2 in humans to accomplish this task.  So as not to confuse the reader, we removed the box on the lower right of the figure related to BCO SNPs (see new Figure 1) and edited the figure footnote to indicate this lack of knowledge.

3. Some linguistic improvement in Table 1.
We have made several changes in Table 1.

Reviewer 2 Report

In my opinion the review is suitable for publication in the Antioxidants journal in its present form, considering that the review is properly organized and  reports updated research work on lycopene digestion, absorption, metabolism and excretion; moreover it also reports on the formation and bioactivity of the lycopene metabolites and their related beneficial health properties.

Author Response

No suggestions made for revision.

Reviewer 3 Report

This review contributes to support the on-going activities of several research groups working on the pair carotenoids/health, with a clear description of the current gaps in knowledge for lycopene attributes (digestion, absorption, metabolism...), some of them also fit to other significant dietary carotenoids (ß-carotin and lutein). There is a substantial revision of the key published literature in this field, with specific contributions of the authors to both research and vision of the available knowledge. There is an interesting section regarding bioactivity of metabolites and their relative bioactive potential considering accumulated amount. I recommend the publication of the article as it is.

Author Response

(The authors gave the same response as above.)
